# More Tolerant Reconstructed Networks Using Self-Healing against Attacks in Saving Resource

**DOI:** 10.3390/e23010102

**Published:** 2021-01-12

**Authors:** Yukio Hayashi, Atsushi Tanaka, Jun Matsukubo

**Affiliations:** 1Graduate School of Advanced Science and Technology/Division of Transdisciplinary Sciences, Japan Advanced Institute of Science and Technology, Nomi 923-1292, Japan; 2Graduate School of Science and Engineering, Yamagata University, Yonezawa 992-8510, Japan; tanaka@yamagata-u.ac.jp; 3Department of Creative Engineering, National Institute of Technology Kitakyushu College, Kitakyushu 802-0985, Japan; jmatsu@apps.kct.ac.jp

**Keywords:** self-healing, network science, resource allocation, enhancing loops, belief propagation, robustness of connectivity, efficiency of paths, resilience

## Abstract

Complex network infrastructure systems for power supply, communication, and transportation support our economic and social activities; however, they are extremely vulnerable to frequently increasing large disasters or attacks. Thus, the reconstruction of a damaged network is more advisable than an empirically performed recovery of the original vulnerable one. To reconstruct a sustainable network, we focus on enhancing loops so that they are not trees, which is made possible by node removal. Although this optimization corresponds with an intractable combinatorial problem, we propose self-healing methods based on enhancing loops when applying an approximate calculation inspired by statistical physics. We show that both higher robustness and efficiency are obtained in our proposed methods by saving the resources of links and ports when compared to ones in conventional healing methods. Moreover, the reconstructed network can become more tolerant than the original when some damaged links are reusable or compensated for as an investment of resource. These results present the potential of network reconstruction using self-healing with adaptive capacity in terms of resilience.

## 1. Introduction

Unfortunately, the frequency of large disasters or malicious attacks is increasing day by day due to climate change, crustal movements, military conflicts, cyber-terrorism, and mega-urbanization. For example, it is well known that a small accident involved a large-area power collapse in North America [1] and the Italian peninsula [2] in 2003, and that an enormous destruction of social infrastructure systems was caused by the great earthquake in Japan in 2011 [3]. There exists a surprisingly common topological structure called scale-free (SF) in many real networks [4], such as power-grids, airlines, communication, transportation systems, and so on, which support social activities, the economy, industrial production, etc. The SF structure is considered to be generated by a selfish rule—preferential attachment [5]—and consists of many low-degree nodes and a few (high-degree) hubs, heterogeneously. Here, degree means the number of links at a node. Moreover, because of the heterogeneity, a SF network has extreme vulnerability against hub attacks [6]. These vulnerable infrastructures appear everywhere and are interdependent on each other. Since a node prefers to connect high-degree nodes in the efficiency bias to shorten the number of path lengths, counted in hops, preferential attachment encourages the heavy concentration of links to hubs. In many real networks, once hubs are damaged and removed due to malfunction, the remaining nodes are fragmented, and lose their basic function for, e.g., communication or transportation. It is a plausible scenario for network infrastructures that the weak points of hubs are involved in large disasters.

Therefore, when large-scale failures or attacks occur, the recovery of the original vulnerable network is inadvisable. Instead, reconstruction by healing is required. By changing the structure instead of recovering the original one, a question arises as to how a sustainable network should be reconstructed to maintain the network function. However, because the linking resources (wire cables, wireless communication, or transportation lines between two nodes, etc.) and ports (channels or plug sockets at a node, etc.) are usually limited, allocation should be controlled at the same time as the rewiring or making additional investment for healing. Such a reconstruction conforms with the concept of resilience in system engineering or ecology as a new supple approach to sustaining basic objectives and integrity, even when encountering extreme changes in situations or environments (e.g., following disasters or malicious attacks) for technological systems, organizations, or individuals [7,8,9].

In this paper, through numerical simulation, we study how to reconstruct a sustainable network under limited-resource conditions, and propose effective self-healing methods based on enhancing loops through a local process around damaged parts. In addition, we show a significant improvement from the previous study [10] by reducing the additional ports prepared in advance besides reusable ports. The motivations for enhancing loops are as follows. In percolation analysis, as a part of network science, it has been found that onion-like structures with positive degree–degree correlations give optimal robustness of connectivity even for a SF network with a power-law degree distribution [11,12]. The “onion-like” description comes from it being visualized by correlations when similar degree nodes are set on a concentric circle arranged in decreasing order of degrees from core to periphery. An onion-like structure can be generated by complete rewiring [11,13], enhancing the correlations under a given degree distribution. On the other hand, since dismantling and decycling problems are asymptotically equivalent at infinite graphs in a large class of random networks with light-tailed degree distribution [14], trees remain without loops at the critical state before complete fragmentation by node removals. The dismantling (or decycling) problem known as NP-hard [15] is aimed at finding the minimum set of nodes for which removal leaves a graph broken into connected components and whose maximum size is at most a constant (or a graph without loops). It is suggested from the equivalence that robustness becomes stronger when as many loops exist as possible. In fact, to be the optimal onion-like network at the same level as the rewired ones [13], enhancing loops by copying [16] or intermediation [17,18] is effective for improving robustness in incrementally growing methods based on a local distributed process such as self-organization. A similar effect is also obtained by preserving or non-preserving the degrees of the nodes after another rewiring based on enhancing loops instead of correlations [19]. Thus, we remark that loops make bypasses, and may be more important than degree–degree correlations improving the connectivity in a network reconstruction after a large disaster or attack. This is predicted to be the top priority for maximizing the decycling set (called the Feedback Vertex Set (FVS) in computer science [15]) so as not to create trees without loops as much as possible even in the worst case of node removal. In other words, enhancing loops corresponds to the optimizing of the tolerance of connectivity in graphs (but not in the context of general computing or problem-solving). Of course, increasing the path lengths between nodes and wasteful resources should be avoided during reconstruction by healing. However, even the identification of the necessary nodes to form loops is intractable due to combinatorial NP-hardness [15]. We effectively apply an approximate calculation by Belief Propagation (BP) based on a statistical physics approach in our self-healing through rewiring (or additional investment), which will be mentioned later. We describe the healing methods as sequential processes for computer simulation, envisioning the further development of distributed control algorithms.

## 2. Methods

### 2.1. Outline of Healing Process

Almost simultaneously attacked nodes are not recoverable immediately. Therefore, they are removed from network function for an amount of time. In such a case of emergency for healing, two unconnected nodes are chosen and rewired, as the reconstruction assistance or reuse of links emanates from the removal of 
qN
 nodes, when the fraction of attacks is *q* and *N* denotes the total number of nodes (as the network size). Some of disconnected links may be reusable on neighboring sides depending on the damage level. Although we call the reuse “rewiring”, removal of nodes is a different problem to that in the so-called “usual” rewiring methods [11,13,19]. The outline of the healing process is as follows.
**Step 0:** Detection and initiationAfter detecting a removal as a malfunction at the nearest neighbor of the attacked node, the healing process is initiated autonomously.**Step 1:** Selection of two nodesSince the neighbor loses links at least temporarily before rewiring, the damaged ones become an attached candidate for healing. Thus, two unconnected nodes are chosen from the neighbors of the nodes removed by attacks. The selections are different in our proposed and the conventional healing methods. Moreover, neighbors are extended in our proposed methods.**Step 2:** Rewiring for healingThe chosen two nodes are connected as rewiring for healing. The above process is repeated for 
$M_{h}\overset{\mathrm{def}}{=}r_{h}\times {\tilde{\sum }}_{i\in D_{q}}k_{i}$
 links.
Here, 
${\tilde{\sum }}_{i\in D_{q}}k_{i}$
 means the number of disconnected links by attacks without multiple counts. 
$D_{q}$
 denotes the set of removed nodes, 
$|D_{q}|\;=\;qN$
. 
$M_{h}$
 includes the number of reused and additionally invested links. When reusable links are insufficient, we assume the addition of links as an investment in the considered 
$M_{h}$
 for a parameter 
$0<r_{h}\leq 1$
 in computer simulation.

In the healing process, rewirings are performed by changing directions and ranges of flight routes or wireless beams. However, we will not discuss the details of realization, which depends on current or future technologies and target systems. Here, we focus on connectivity at the most fundamental level in many network systems for not only communication but also transportation, power supply, and other infrastructures, while our approach may be useful for path control or failure detection, e.g., by software-defined network-based reconfiguration of communication systems with switches in managing reliability, latency, or security at some service levels [20,21,22]. In addition, we consider that ports work independently from links, similar to the relationship between an airport runaway or plug socket and an airplane flight or cable line. It is a reasonable assumption that the amount of degree 
$k_{j}$
 ports is reusable at the undamaged neighbor node 
j∈∂i
 of a removed node *i* by attacks, where 
∂i
 denotes a set of the nearest connecting neighbors of *i*. Thus, there exist at least as many active (reusable) ports of a node as its degree in the original network before any attacks.

### 2.2. Proposed Healing Methods

In our proposed healing methods, there are two phases: ring formation and enhancing loops by applying BP in the next subsection. Moreover, they (RingRecal, RingLimit1,5,10, RingLimit5Recal) are modified to reduce the additional ports from the previous results [10] by avoiding the concentration of links at some nodes.
**RingBP** Previously, our combination method of ring formation and enhancing loops was based on [10]. After making rings on the extended neighbors of removed nodes, as shown in Figure 1, enhancing the loops on the rings is performed by applying the BP algorithm [23] (see Section 2.3). However, in BP, a set 
$\left\{p_{i}^{0}\right\}$
 as probability of node *i* to be necessary for loops is calculated only once, just after an attack. Please note that a ring is the simplest loop when it uses the least number of links.**RingRecal** We modified our method with the recalculation of BP. After making rings, a set 
$\left\{p_{i}^{0}\right\}$
 is recalculated one by one through each rewiring in the remaining links within 
$M_{h}$
 for enhancing loops.**RingLimit1,5,10** We modified our method with limited rewirings. After making rings, in enhancing loops, the number of rewiring links is limited at node *i* to its degree 
$k_{i}$


+1
, 
+5
, or 
+10
.**RingLimit5Recal** We modified our method by a combination of RingRecal and RingLimit5. After making rings, a set of 
$\left\{p_{i}^{0}\right\}$
 is recalculated one by one through each rewiring in the remaining links within 
$M_{h}$
 for enhancing loops. Moreover, the number of rewiring links is limited at node *i* to 
$k_{i}+5$
.

First, in ring formation (see Figure 1), the order of process is basically according to the order of the removed nodes 
$i_{1},i_{2},\ldots ,i_{qN}$
. Thus, rings are made for the neighbors 
$\partial i_{1},\partial i_{2},\ldots ,\partial i_{qN}$
 in this order. However, if there is 
$i_{k^{\prime }}\in \partial i_{k}$
, 
$k^{\prime }>k$
, it is extended as the union 
$\partial i_{k}\leftarrow \partial i_{k}\cup \partial i_{k^{\prime }}$
. In addition, if there is 
$i_{k^{\prime \prime }}\in \partial i_{k^{\prime }}$
, 
$k^{\prime \prime }>k^{\prime }$
, it is also extended as the union 
$\partial i_{k}\leftarrow \partial i_{k}\cup \partial i_{k^{\prime }}\cup \partial i_{k^{\prime \prime }}$
. Such extensions of neighbors are repeated until a ring encloses the induced subgraph of removed nodes and their links. To make a ring, a node is chosen u.a.r. and connected to a subsequent similarly chosen node in a set of the extended neighbors. This is repeated without multi-selections until a return is made to the first chosen node from the last chosen node.

Next, when enhancing loops on each ring for the remaining rewirings in 
$M_{h}$
, a node *j* with the minimum 
$p_{j}^{0}$
 is chosen in all the neighbors of the removed nodes, and connected to the other node 
$j^{\prime }$
 with the second minimum 
$p_{j^{\prime }}^{0}$
 on the ring to which *j* belongs. For each rewiring, a set of 
$\left\{p_{i}^{0}\right\}$
 is recalculated one by one in RingRecal and RingLimit5Recal methods. In addition, the number of rewiring links is limited at node *i* to 
$k_{i}+5$
 (or 
+1
, 
+10
) according to its degree 
$k_{i}$
 in RingLimit5Recal and RingLimit5 (or RingLimit1, RingLimit10) methods. If the condition is unsatisfied, another node with a second, third, fourth, and subsequent minimum is chosen as a candidate for healing. Although a node 
$j^{\prime \prime }$
 with small 
$p_{j^{\prime \prime }}^{0}$
 tends not to contribute to making loops because it is not included in FVS, it is expected that the number of loops is increased by connecting such nodes. This is the reason for the above selection.

### 2.3. Applying the Belief Propagation Algorithm

To calculate the probability 
$p_{i}^{0}$
 of belonging to FVS, the following BP algorithm [23] is applied. It is based on a cavity method in statistical physics. We review the outline derived for approximately estimating FVS known as the NP-hard problem [15]. In the cavity graph, it is assumed that nodes 
j∈∂i
 are mutually independent of each other when node *i* is removed (the exception is denoted by 
\i
). Then the joint probability is 
$P_{\backslash i}\left(A_{j}:j\in \partial i\right)\approx \Pi_{j\in \partial i}p_{j\to i}^{A_{j}}$
 by the product of independent marginal probability 
$p_{j\to i}^{A_{j}}$
 for the state 
$A_{j}$
 as the node index of *j*’s root or empty 0: it belongs to FVS. The corresponding probabilities are represented by

(1)
$$p_{i}^{0}\overset{\mathrm{def}}{=}\frac{1}{z_{i}\left(t\right)},$$


(2)
$$p_{i\to j}^{0}=\frac{1}{z_{i\to j}\left(t\right)},$$


(3)
$$p_{i\to j}^{i}=\frac{e^{x}\Pi_{k\in \partial i(t)\backslash j}\left[p_{k\to i}^{0}+p_{k\to i}^{k}\right]}{z_{i\to j}\left(t\right)},$$

where 
∂i(t)
 denotes node *i*’s set of connecting neighbor nodes at time *t*, and 
x>0
 is a parameter of inverse temperature. The normalization constants are

(4)
$$z_{i}\left(t\right)\overset{\mathrm{def}}{=}1+e^{x}\left[1+\sum_{k\in \partial i(t)}\frac{1-p_{k\to i}^{0}}{p_{k\to i}^{0}+p_{k\to i}^{k}}\right]\Pi_{j\in \partial i(t)}\left[p_{j\to i}^{0}+p_{j\to i}^{j}\right],$$


(5)
$$\begin{array}{ccc} z_{i\to j}\left(t\right) & \overset{\mathrm{def}}{=} & 1+e^{x}\Pi_{k\in \partial i(t)\backslash j}\left[p_{k\to i}^{0}+p_{k\to i}^{k}\right]\times \left[1+\sum_{l\in \partial i(t)\backslash j}\frac{1-p_{l\to i}^{0}}{p_{l\to i}^{0}+p_{l\to i}^{l}}\right], \end{array}$$

to be satisfied for any node *i* and link 
i→j
 as

$$p_{i}^{0}+p_{i}^{i}+\sum_{k\in \partial i}p_{i}^{k}=1,\;\;\;p_{i\to j}^{0}+p_{i\to j}^{i}+\sum_{k\in \partial i}p_{i\to j}^{k}=1.$$

We repeat these calculations of message-passing until they are self-consistent in principle but practically reach appropriate rounds from an initial setting of 
(0,1)
 random values. The unit time from *t* to 
t+1
 for calculating a set 
$\left\{p_{i}^{0}\right\}$
 consists of several rounds by updating Equations (Equation 1)–(Equation 5) in order of random permutation of the total *N* nodes. Since the sums or products in Equations (Equation 1)–(Equation 5) are restricted in the nearest neighbor, they are local processes. The distributed calculations can be also considered. As included in FVS, a node *k* with the maximum 
$p_{k}^{0}$
 is chosen. After removing the chosen node, 
$\left\{p_{i}^{0}\right\}$
 is recalculated the next time. Such a process is repeated until it is acyclic for finding the FVS. However, in our healing method, 
$\left\{p_{i}^{0}\right\}$
 is used to select the attached two nodes on a ring by rewiring.

### 2.4. Conventional Healing Methods

We briefly explain the following typical healing methods in network science (inspired by fractal statistical physics) and computer science.
**RBR** Conventional Random Bypass Rewiring (RBR) method [24] (corresponds to 
$r_{h}=0.5$
).**GBR** Greedy Bypass Rewiring (GBR) method improved from RBR heuristically [24].**SLR** Conventional Simple Local Repair (SLR) method [25] with priority of rewirings to more damaged nodes.

In network science, a self-healing method that adds new random links to interdependent two-layered networks of square lattices has been proposed, and the effect against node attacks is numerically studied [26]. In particular, for the addition of links by the healing process, the candidates of linked nodes are incrementally extended from only the direct (nearest connecting) neighbors of the removed node by attacks until no more separation of components occurs. In other words, connectivity is maintained except for the isolating removed parts. Such extension of the candidates of linked nodes is a key idea in our proposed self-healing method.

Furthermore, the following self-healing methods, whose effects are investigated for some data of real networks, are worthy to note. One is a distributed SLR [25] with repair by a link between the most damaged node and a randomly chosen node from the unremoved node set in its next-nearest neighbor before attacks. The priority of the damaged nodes is according to the smaller fraction 
$k_{dam}/k_{orig}$
 of its remained degree 
$k_{dam}$
 and the original degree 
$k_{orig}$
 before the attacks. The selections are repeated until reaching a given rate 
$f_{s}$
 controlled by the fraction of nodes whose 
$k_{dam}/k_{orig}$
 falls below a threshold. Another is RBR [24] on the more limited resources of links and ports. To establish links between pair nodes, a node is randomly chosen only one time from the neighbors of each removed node. When 
$k_{i}$
 denotes the degree of the removed node *i*, only 
$\left\lfloor k_{i}/2\right\rfloor$
 links are reused. Please note that reserved additional ports are not necessary: they do not exceed the original one before attacks. Moreover, GBR [24] is proposed to improve the robustness; the selection of pair nodes is based on the number of the links not yet rewired and the size of the neighboring components.

In computer science, the ForgivingTree algorithm has been proposed [27]. Under repeated attacks, the following self-healing is processed one by one after each node removal, except when the removed node is a leaf (whose degree is one). This is based on both a distributed process of sending messages and data structure, further developed to an efficient algorithm called compact routing [28]. In each rewiring process, a removed node and its links are replaced by a binary tree. Please note that each vertex of the binary tree was the neighbors of the removed node, whose links to the neighbors are reused as the edges of the binary tree. Thus, additional links for healing are unnecessary, but not controllable. It is remarkable for computation (e.g., in routing or information spreading) that the multiplicative factor of the diameter of the graph after healing is never more than 
$O(\log k_{max})$
, where 
$k_{max}$
 is the maximum degree in the original network because of the replacement by binary trees. However, the robustness of connectivity is not taken into account in the limited rewiring based on binary trees, since a tree structure is easily fragmented into subtrees by any attack on the articulation node. Thus, this healing method is excluded from compared ones.

## 3. Results

We evaluate the effect of healing by four measures: the ratio 
$S\left(q\right)/N_{q}$
 [25] for the connectivity, the robustness index 
$R\left(q\right)\overset{\mathrm{def}}{=}\sum_{q^{\prime }}S_{q}\left(q^{\prime }\right)/N_{q}$
, the efficiency of paths 
$E\left(q\right)\overset{\mathrm{def}}{=}\frac{1}{N_{q}\left(N_{q}-1\right)}\sum_{i\neq j}\frac{1}{L_{ij}}$
, and the average degree 
$k_{avg}\left(q\right)$
 in 
$N_{q}\overset{\mathrm{def}}{=}\left(1-q\right)N$
 nodes, where 
S(q)
 and 
$S_{q}\left(q^{\prime }\right)$
 denote the sizes of GC (giant component or largest connected cluster) after removing 
qN
 nodes by attacks from the original network and removing 
$q^{\prime }N_{q}$
 nodes by further attacks from the surviving 
$N_{q}$
 nodes, respectively. Here, a removed node is chosen with recalculation of the highest degree node as the target. Remember that 
q=1/N,2/N,…,(N−1)/N
 (or 
$q^{\prime }=1/N_{q},2/N_{q},\ldots ,\left(N_{q}-1\right)/N_{q}$
) is a fraction of attacks. Although 
S(q)
 or 
$S_{q}\left(q^{\prime }\right)$
 represents the size of GC after attacks to 
qN
 or 
$q^{\prime }N_{q}$
 nodes, 
R(q)
 is a measure of tolerance of connectivity against further attacks. 
$L_{ij}$
 denotes the length of the shortest path counted by hops between *i*-*j* nodes in the surviving 
$N_{q}$
 nodes. The ranges are 
$0<S\left(q\right)/N_{q}\leq 1$
, 
0<R(q)≤0.5
, and 
0<E(q)≤1
. We investigate the four measures before or after healing for OpenFlights between airports, Internet AS Oregon, and US PowerGrid as examples of typical infrastructures of SF networks [29,30,31] after extracting from each of them to a connected and undirected graph without multiple links (see Table 1). We compare the results shown by color lines with marks in figures for the conventional RBR, GBR, SLR, and our proposed RingBP, RingRecal, RingLimit1,5,10, RingLimit5Recal methods.

In each of Figure 2, Figure 3, Figure 4 and Figure 5, no-healing, conventional, and previous our methods are compared in (a), previous our and RingRecal or RingLimit methods are compared in (c,d), previous our and the best combination RingLimit5Recal methods are compared in (b). Red, green, blue, orange, and purple lines denote the rate 
$r_{h}=0.05,0.1,0.2,0.5$
 and 
1.0
, respectively, for the number 
$M_{h}$
 of rewirings. The results for the original and no-healing networks are shown by dashed magenta and solid black lines. The following results are averaged over 100 samples with random process for tie-breaking in a node selection or ordering of nodes on a ring.

Figure 2 shows the ratio 
$S\left(q\right)/N_{q}$
 of connectivity in the surviving 
$N_{q}$
 nodes. Remember that 
S(q)
 is the size of GC after healing (or no-healing) against attacks to 
qN
 nodes. Higher ratio means larger connectivity, as maintaining the network function for communication or transportation, 
$S\left(q\right)/N_{q}<1$
 indicates the incomplete ring formation stopped in 
$M_{h}$
. As shown in Figure 2a, the ratio rapidly decreases in the conventional SLR method marked by open circles for OpenFlights and PowerGrid, while it is moderately higher around 
$S\left(q\right)/N_{q}\approx 0.5$
 on purple and blue lines or increasing in green and red lines marked by open circles for AS Oregon. Moreover, in Figure 2a, the following results are common for OpenFlights, AS Oregon, and PowerGrid. The ratio also decreases in the conventional RBR and GBR methods denoted by dashed light blue and brown lines, respectively. In the corresponding RingBP method, the ratio is the highest as the horizontal orange (but overlapped purple) line at 
$S\left(q\right)/N_{q}\approx 1.0$
 marked by open squares. The bottom dashed black lines around 
$S\left(q\right)/N_{q}=0$
 are the results without the network function for no-healing. Thus, previously, our RingBP method marked by open squares had higher ratio than the conventional methods in comparison with the same color lines. Figure 2b,c shows that the ratio in RingBP method marked by open squares almost coincides with the ones in the RingLimit5Recal method marked by open diamonds and RingRecal method marked by filled diamonds. Similarly, Figure 2d shows that the ratio in RingBP method marked by open squares almost coincides with ones in RingLimit5,10 methods marked by lower triangles and asterisks. However, it is slightly lower in the RingLimit1 method marked by open upper triangles. Therefore, RingLimit5Recal, RingRecal, and RingLimit5,10 methods are the best at the same level to RingBP in maintaining connectivity. The constraint to the number of additional ports is slightly too strong as there is only one in the RingLimit1 method.

Figure 3 shows the robustness index 
R(q)
 as the tolerance of connectivity against further attacks to the surviving 
$N_{q}$
 nodes after healing. Please note that a major part of 
$N_{q}$
 nodes belong to the GC, but other parts belong to isolated clusters. In Figure 3a for OpenFlights, AS Oregon, and PowerGrid, the values of 
R(q)
 rapidly decrease to very low level 
≤0.1
 with vulnerability in the conventional SLR method marked by open circles and in RBR and GBR methods denoted by light blue and brown dashed lines, while there exist higher values of 
R(q)
 (on purple and orange lines for 
$r_{h}\geq 0.5$
) in RingBP method marked by open squares than the horizontal dashed magenta lines in the original network. The results for no-healing are at the bottom as 
R(q)≈0
 because of 
$S_{q}\approx 0$
 from Figure 2a. Moreover, as shown in Figure 3b,c, RingLimit5Racal method marked by open diamonds and RingRecal method marked by filled diamonds have higher values of 
R(q)
 than RingBP marked by open squares in comparison with same color lines for OpenFlights and AS Oregon, while these methods have almost same values of 
R(q)
 to ones in RingBP for PowerGrid. Similarly, as shown in Figure 3d for OpenFlights and AS Oregon, RingLimit1,5,10 methods marked by open lower, upper triangles, and asterisks have higher values of 
R(q)
 than RingBp method marked by open squares in comparison with same color lines. However, the difference becomes smaller in green and red lines for 
$r_{h}\leq 0.1$
. In Figure 3d for PowerGrid, similar values of 
R(q)
 are obtained on each color line regardless of marks for different methods. In particular, for OpenFlights and AS Oregon, purple, orange and blue line (
$r_{h}\geq 0.2$
) in RingLimit5Recal marked by open diamonds are slightly higher than ones in Ringlimit1,5,10 marked by open upper, lower triangles and asterisks as shown in Figure 3b,d. Thus, the reconstructed networks by our proposed healing methods can become stronger with higher values of 
R(q)
 than the original network against further attacks. In particular, the improvement is remarkable from 
R(q)<0.1
 to 
R(q)>0.3
 for OpenFlights and AS Oregon.

Figure 4 shows the efficiency 
E(q)
 of shortest paths between two nodes in the surviving 
$N_{q}$
 nodes. Please note that 
E(q)=0.1,0.2,0.25
 corresponds to 
10,5,4
 hops of the average path length 
$L_{avg}\left(q\right)$
 from 
$L_{avg}\left(q\right)\approx 1/E\left(q\right)$
 in the arithmetic and the harmonic means of path lengths. The following results are common for OpenFlights, AS Oregon, and PowerGrid. As with Figure 2a–Figure 4a that show that the values of 
E(q)
 rapidly decrease in the conventional SLR method marked by open circles, RBR and GBR methods are denoted by light blue and brown dashed lines, while the values are higher in RingBP method are marked by squares in comparison with same color lines. In Figure 4b,c, RingLimit5Recal method marked by open diamonds and RingRecal method marked by filled diamonds have similar or slightly lower values of 
E(q)
 than ones in RingBP method marked by squares in comparison with the same color lines. In Figure 4d for OpenFlights and AS Oregon, the values are slightly lower in RingLimit1,3,5 methods marked by open upper triangles, lower triangles, and asterisks than ones in RingBP method marked by squares, while for PowerGrid the values are similar regardless of these methods in comparison with same color lines.

Figure 5 shows the average degree 
$k_{avg}\left(q\right)$
 in the surviving 
$N_{q}$
 nodes. This value indicates how much links are effectively used for hearing. In other words, a small value of 
$k_{avg}\left(q\right)$
 means that rewirings are restricted and not fully used until the possible number 
$M_{h}$
 of links, especially in conventional methods, by the constraints on linking between not the extended but the nearest neighbors of attacked nodes or the limitation (see Section 2.4). The following results are common for OpenFlights, AS Oregon, and PowerGrid. As shown in Figure 5a, it is remarkable that the values are small (
$k_{avg}\left(q\right)<10$
) in the conventional SLR method marked by open circles, and RBR and GBR methods denoted by dashed light blue and brown lines, while the values are higher in RingBP method marked by open squares, in comparison with the same color lines. In Figure 5b,d, by saving rewired links due to the limitation of additional ports, the values of 
$k_{avg}\left(q\right)$
 are not large in the RingLimit5Recal method marked by open diamonds or in RingLimit1,5,10 methods marked by open upper triangles, lower triangles and asterisks. In Figure 5c, on each color line, the values of 
$k_{avg}\left(q\right)$
 in RingBP method marked by open squares are almost coincident with the ones in the RingRecal method marked by filled diamonds. However, in RingLimit5Recal and RingLimit1,5,10 methods with saving rewired links, both 
R(q)
 and 
E(q)
 are high values as shown in Figure 3b,d and Figure 4b,d. Therefore, these methods are more effective for healing to improve the robustness and efficiency to similar levels by using less resource. Figure 6 shows that the reconstructed degree distribution 
P(k)
 in RingLmit5Recal method becomes exponential approximately in a semi-logarithmic plot from a power-law in the original network. The maximum degrees are bounded as 65, 19, and 14 for OpenFlights, AS Oregon, and PowerGrid, respectively. They tend to be smaller as *q* increases.

Table 2 shows the maximum number (or in parentheses, the average value over the nodes that perform much more rewirings than their degrees of the reusable number of ports) of additional ports in the RingRecal method. Although the values are reduced to less than 
$k_{max}$
 from nearly 
$2k_{max}∼3k_{max}$
 in our previous RingBP method [10], they are still large. Here, 
$k_{max}$
 is 242, 1458, or 19 for the original networks OpenFlights, AS Oregon, or PowerGrid are shown in Table 1. Of course, the maximum number of additional ports is significantly restricted to a constant 
1,5,10
 or 5 in RingLimit1,5,10 or RingLimit5Recal method. Since the additional ports should be stored in advance beside a reusable number of its degree in the original network, less preparation is better with a lower investment cost of resource. Thus, RingBP or RingRecal method is not desirable, because it requires many additional ports.

## 4. Discussion

We have proposed self-healing methods with modifications from the previous one [10] for reconstructing a resilient network through rewirings against attacks or disasters in the resource allocation control of links and ports. The healing strategy is based on maintaining connectivity using ring formation on the extended neighbors of attacked nodes and enhancing loops to improve the robustness of connectivity in applying the approximate calculations of BP [23] inspired by statistical physics in a distributed manner. We have taken into account the limitation of additions and the recalculations of BP as modifications to reduce the preparation of additional ports by avoiding the concentration of links at some nodes.

Simulation results show that our proposed combination methods of ring formation and enhancing loops are better than the conventional SLR [25], RBR, and GBR [24] methods. In particular, in the RingLimit5Recal method, both high robustness of connectivity and efficiency of paths are obtained by saving the resource of links and ports, even though the number of additional ports is significantly restricted to a constant 5 from the previous 
$O\left(k_{max}\right)∼10^{3}$
 [10]. Moreover, we have found that the reconstructed networks by healing can become more robust and efficient than the original networks before attacks, when some extent of damaged links are reusable or compensated for as the rate 
$r_{h}\geq 0.5$
.

However, what structure is the optimally tolerant against further attacks in varying the degree distribution after healing remains an open question. Even if our prediction comes true, it is not yet known what approach is more effective and practical for approximately maximizing the FVS. These challenging problems are beyond the discussion of onion-like structure under a given degree distribution [11,12]. In addition, how the healing method should be extended to interdependent or multilayer networks as networks of networks is an intensive issue. On the other hand, in application points of view, further investigations will be useful for other networks if huge computation is available, since our obtained results seem to depend on special topological structures such as PowerGrid with a large diameter *D* (see Table 1). The development of distributed algorithms within only local information is also important for our self-healing methods.

## Figures and Tables

**Figure 1 entropy-23-00102-f001:**
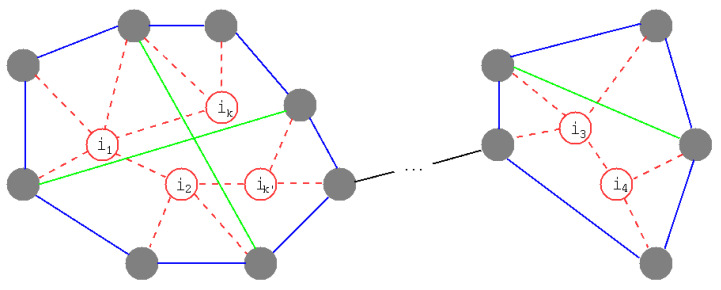
Schematic illustration of ring formation and enhancing loops. Red nodes and their links are removed by attacks. Gray filled nodes are the neighbors. Blue lines make rings, and green lines are rewirings for enhancing loops on rings.

**Figure 2 entropy-23-00102-f002:**
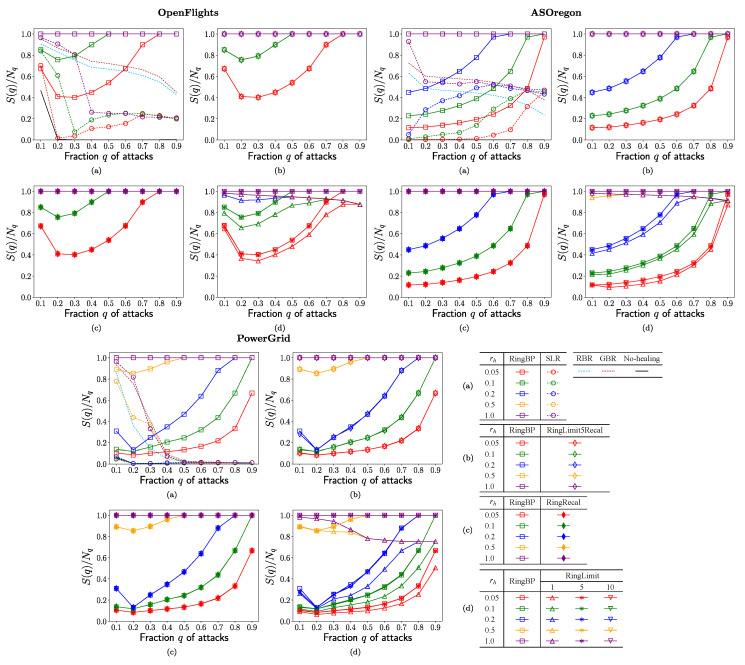
Ratio 
$S\left(q\right)/N_{q}$
 of connectivity vs. fraction *q* of attacks for the rate 
$r_{h}$
 in rewirings. (**a**) SLR, RBR, GBR, RingBP, (**b**) RingBP, RingLimit5Recal, (**c**) RingBP, RingRecal, and (**d**) RingBP, RingLimit1,5,10 methods.

**Figure 3 entropy-23-00102-f003:**
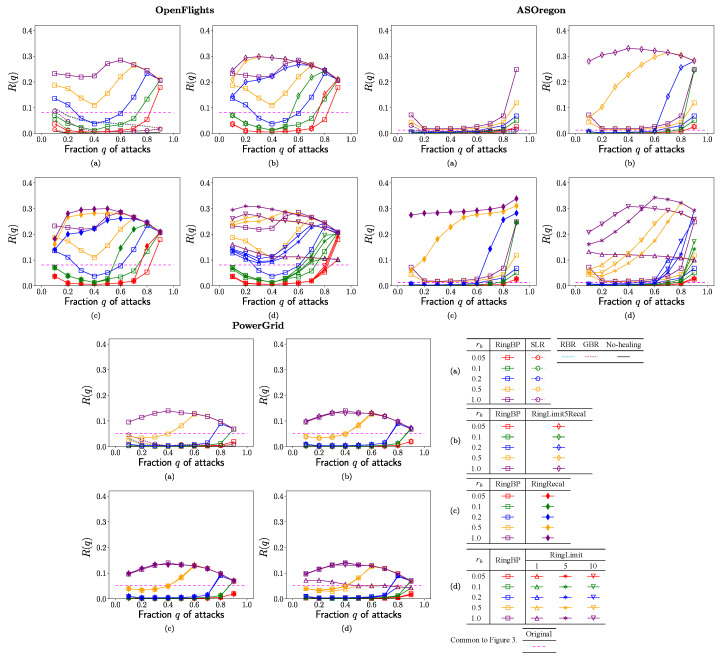
Robustness index as the tolerance of connectivity against further attacks to the surviving 
$N_{q}$
 nodes vs. fraction *q* of attacks for the rate 
$r_{h}$
 in rewirings. (**a**) SLR, RBR, GBR, RingBP, (**b**) RingBP, RingLimit5Recal, (**c**) RingBP, RingRecal, and (**d**) RingBP, RingLimit1,5,10 methods.

**Figure 4 entropy-23-00102-f004:**
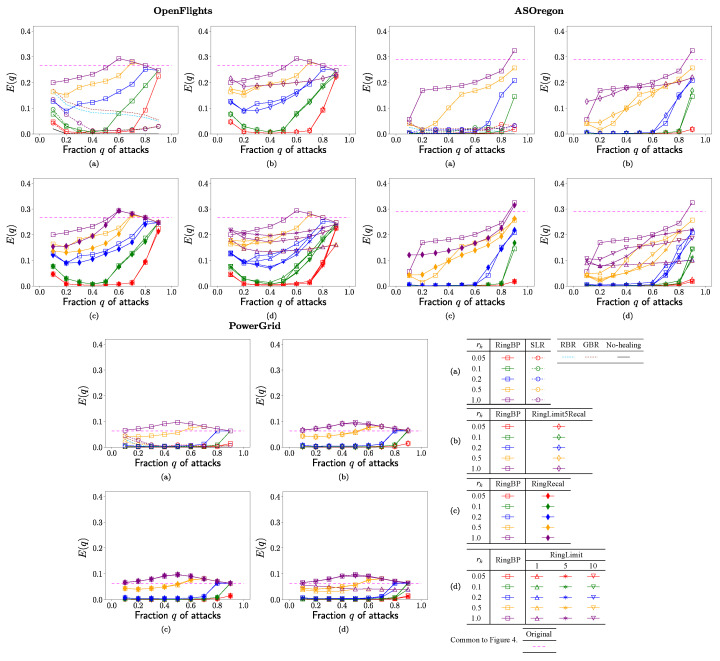
Efficiency of paths in the surviving 
$N_{q}$
 nodes vs. fraction *q* of attacks for the rate 
$r_{h}$
 in rewirings. (**a**) SLR, RBR, GBR, RingBP, (**b**) RingBP, RingLimit5Recal, (**c**) RingBP, RingRecal, and (**d**) RingBP, RingLimit1,5,10 methods.

**Figure 5 entropy-23-00102-f005:**
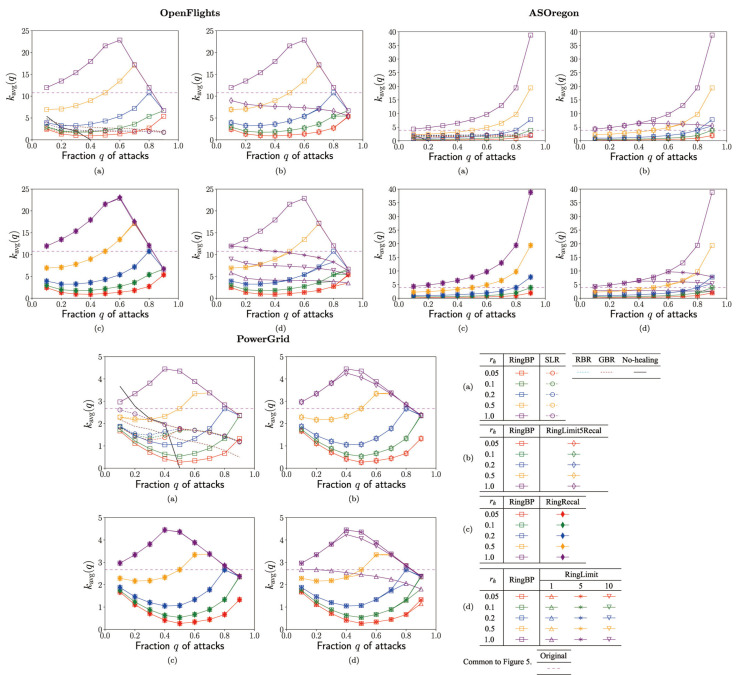
Averagedegree 
$k_{avg}\left(q\right)$
 in the surviving 
$N_{q}$
 nodes vs. fraction *q* of attacks for the rate 
$r_{h}$
 in rewirings. (**a**) SLR, RBR, GBR, RingBP, (**b**) RingBP, RingLimit5Recal, (**c**) RingBP, RingRecal, and (**d**) RingBP, RingLimit1,5,10 methods.

**Figure 6 entropy-23-00102-f006:**
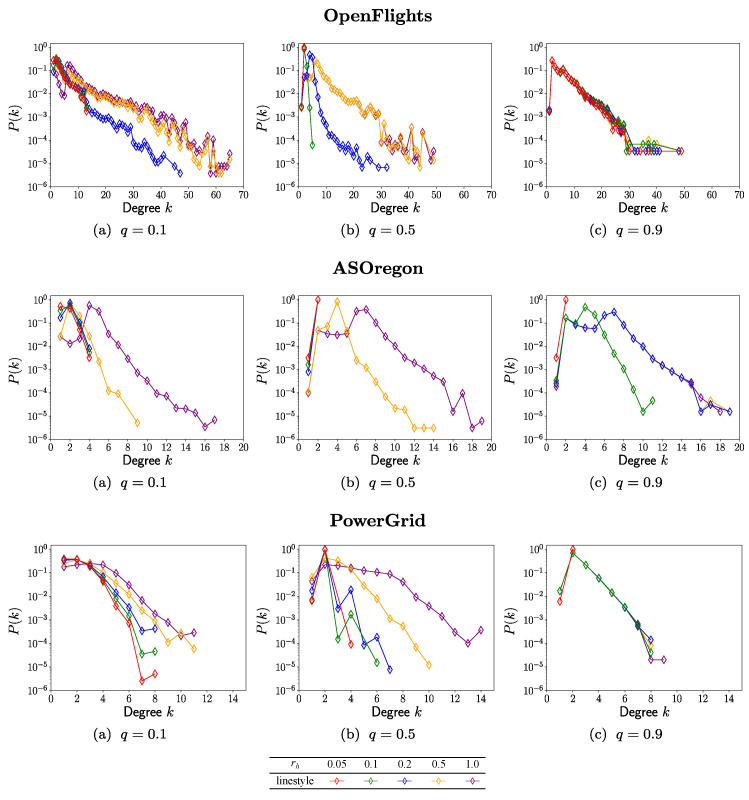
Degree distribution in surviving 
$N_{q}$
 nodes after healing by RingLimit5Recal method for the fraction of attacks.

**Table 1 entropy-23-00102-t001:** Basic properties for the original networks. *N* and *M* denote the numbers of nodes and links. 
$k_{avg}=2M/N$
, 
$k_{min}$
, and 
$k_{max}$
 are the average, minimum, and maximum degrees. 
$L_{avg},D,R$
 and *E* denote the average path length, diameter, robustness index, and efficiency of paths, respectively.

Network	*N*	*M*	$k_{\mathrm{avg}}$	$k_{\min}$	$k_{\max}$	$L_{\mathrm{avg}}$	*D*	*R*	*E*
OpenFlight	2905	15,645	10.77	1	242	4.097	14	0.080912	0.266934
AS Oregon	6474	12,572	3.883	1	1458	3.705	9	0.012500	0.290399
PowerGrid	4941	6594	2.669	1	19	18.989	46	0.052428	0.062878

**Table 2 entropy-23-00102-t002:** Maximum number of additional ports (the average number in parenthesis) for the fraction *q* of attacks and the rate 
$r_{h}$
 in rewirings.

	**OpenFlights**								
	0.1	0.2	0.3	0.4	0.5	0.6	0.7	0.8	0.9
0.05	1	1	1	1	1	1	1	2	14.7
	(1)	(1)	(1)	(1)	(1)	(1)	(1)	(1.36)	(3.32)
0.1	1	1	1	1	2	2	4.03	20.68	21.59
	(1)	(1)	(1)	(1)	(1.10)	(1.37)	(1.91)	(3.55)	(4.46)
0.2	26.54	3.67	3.44	3.81	10.22	29.44	44.08	49.23	21.56
	(1.66)	(1.59)	(1.68)	(1.91)	(2.51)	(3.64)	(5.47)	(8.46)	(4.46)
0.3	55.44	23.93	26.55	42.07	50.64	65.58	69.37	63.04	21.61
	(3.57)	(2.83)	(3.02)	(3.72)	(4.86)	(6.56)	(8.92)	(9.74)	(4.46)
0.4	77.65	56.61	62.22	69.58	79.96	88.11	81.62	63.09	21.6
	(5.57)	(4.44)	(4.81)	(5.74)	(7.20)	(9.28)	(12.17)	(9.73)	(4.46)
0.5	96.35	81.88	87.52	98.26	106.62	105.12	107.14	62.99	21.62
	(7.56)	(6.07)	(6.61)	(7.77)	(9.59)	(11.84)	(14.84)	(9.74)	(4.46)
1.0	179.04	175.97	173.53	177.32	160.94	170.93	115.33	63.02	21.57
	(16.78)	(13.95)	(14.85)	(16.91)	(19.68)	(20.66)	(15.10)	(9.74)	(4.46)
	**AS Oregon**								
	0.1	0.2	0.3	0.4	0.5	0.6	0.7	0.8	0.9
0.05	1	1	1	1	1	1	1	1	1
	(1)	(1)	(1)	(1)	(1)	(1)	(1)	(1)	(1)
0.1	1	1	1	1	1	1	1	1	6.43
	(1)	(1)	(1)	(1)	(1)	(1)	(1)	(1)	(2.26853)
0.2	1	1	1	1	1	1	2.03	6.44	56.47
	(1)	(1)	(1)	(1)	(1)	(1)	(1.29)	(2.23)	(6.30)
0.3	1	1	1	1	2	3.08	6.67	49.06	85.74
	(1)	(1)	(1)	(1)	(1.21)	(1.42)	(2.22)	(4.21)	(10.38)
0.4	1	1	2	2	3.66	5.48	48.89	79.8	107.53
	(1)	(1)	(1.16)	(1.31)	(1.57)	(2.21)	(3.53)	(6.24)	(14.48)
0.5	2.49	2	2	3.77	6.22	48.81	79.89	104.12	127.04
	(1.11)	(1.26)	(1.37)	(1.68)	(2.22)	(3.20)	(4.88)	(8.25)	(18.54)
1.0	40.64	73.63	93.29	111.4	128	146.73	164.37	181.31	199.37
	(2.67)	(3.20)	(3.92)	(4.87)	(6.22)	(8.21)	(11.56)	(18.30)	(38.81)
	**PowerGrid**								
	0.1	0.2	0.3	0.4	0.5	0.6	0.7	0.8	0.9
0.05	1.08	1	1.02	1.01	1	1	1	1	1
	(1.00)	(1)	(1.00)	(1.00)	(1)	(1)	(1)	(1)	(1)
0.1	1.65	1.99	1.93	1.59	1.15	1.02	1.04	1.02	3.81
	(1.01)	(1.01)	(1.02)	(1.00)	(1.00)	(1.00)	(1.00)	(1.00)	(1.19)
0.2	2	2.1	2.12	2.25	2.13	1.91	2.01	3.8	3.69
	(1.02)	(1.02)	(1.03)	(1.03)	(1.02)	(1.00)	(1.01)	(1.32)	(1.19)
0.3	2	2.59	2.95	2.93	2.95	3.01	3.12	6.21	3.74
	(1.02)	(1.05)	(1.07)	(1.07)	(1.08)	(1.10)	(1.27)	(1.51)	(1.19)
0.4	2.42	3.06	3.25	3.14	3.25	3.11	8.99	6.27	3.74
	(1.04)	(1.09)	(1.10)	(1.13)	(1.15)	(1.26)	(1.90)	(1.51)	(1.19)
0.5	2.87	3.76	3.47	3.47	3.31	5.66	9.03	6.35	3.77
	(1.06)	(1.11)	(1.14)	(1.17)	(1.27)	(1.78)	(1.90)	(1.51)	(1.19)
1.0	4.9	6.08	13.42	15.44	14.95	11.85	9.01	6.29	3.78
	(1.64)	(1.86)	(2.31)	(2.94)	(2.80)	(2.35)	(1.90)	(1.51)	(1.19)

## Data Availability

Data sharing not applicable.

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
