# Peer review of "More Tolerant Reconstructed Networks Using Self-Healing against Attacks in Saving Resource"

_entropy, 2021, doi:10.3390/e23010102_

Round 1

Reviewer 1 Report

This manuscript provides an interesting approach to reconstruct networks from large-scale attacks.  This paper presents some interesting findings and their relevance is proved by performing a lot of simulation studies corresponding to a widely potential applicative direction. 

One minor comment regards this paper: It is better to move the "Method" section before the "results" section. 

Author Response

We would like to express our thanks for the valuable comments.
In the following sentences, RC# and OA# denote Reviewer's Comment
and Our Answer + sequential number.

For reviewer 1
RC1: One minor comment regards this paper:
It is better to move the "Method" section before the "results" section.
OA1: These sections are exchanged according to the above suggestion in
remaining the subsection 2.1 and moving the brief description list
of healing methods (in lines 118-123 on page 2, lines 124-134 on page 3,
lines 165-168 on page 5). The readability will be improved.

Reviewer 2 Report

Summary: Large networks are susceptible to attacks which is generally solved by combinatorial solutions. Self-healing method proposed using local loops through process around damaged parts. Authors conduct a numerical simulation to reconstruct the network with limited resource bypassing the regular paths of computing using shorter and limited loops.

Comments:

  1. Authors have presented their work in great details, and the results presented are substantial for their claims. However, certain things that would help improve the article are listed below.
  2. Please list the legend for different curves in the figures. They are confusing to capture the relative performances. Also, name the actual parameter of study than a symbol or representative quantity.
  3. What does a loop correspond to in the context of computing and optimization problem solving, as well as in relation to networking?
  4. Where do you envision the optimization algorithm to run in the real deployment? a centralized location, or as a distributed computing?
  5. How does SDN help in solving the problem? Are you considering an SDN based run-time dynamic reconfiguration?
  6. Can you present the algorithms in the form of short flow chart of small steps or an algorithm?
  7. The citations can be improved. Authors can consider the citations in the area of reliability and SDN such as:
    1. Song, Sejun, et al. "Control path management framework for enhancing software-defined network (SDN) reliability." IEEE Transactions on Network and Service Management 14.2 (2017): 302-316.
    2. Hu, Yannan, et al. "Reliability-aware controller placement for software-defined networks." 2013 IFIP/IEEE International Symposium on Integrated Network Management (IM 2013). IEEE, 2013.
    3. Thyagaturu, Akhilesh S., et al. "Software defined optical networks (SDONs): A comprehensive survey." IEEE Communications Surveys & Tutorials 18.4 (2016): 2738-2786.

Author Response

We would like to express our thanks for the valuable comments.
In the following sentences, RC# and OA# denote Reviewer's Comment
and Our Answer + sequential number.

For reviewer 2
RC2: Please list the legend for different curves in the figures.
They are confusing to capture the relative performances.
Also, name the actual parameter of study than a symbol or
representative quantity.
OA2: The legends are added for all relevant figures.

RC3: What does a loop correspond to in the context of computing
and optimization problem solving, as well as in relation to
networking?
OA3: We enhance "In other words, enhancing loops correspond to
optimizing the tolerance of connectivity in graphs (but not
in the contents of general computing or problem solving)."
in lines 70-72 on page 2.

RC4: Where do you envision the optimization algorithm to run
in the real deployment? a centralized location, or as a
distributed computing?
OA4: We explain "We describe the healing methods as sequential
processes for computer simulation in envisioning the further
development of distributed control algorithms." in lines 76-78
on page 2.

RC5: How does SDN help in solving the problem? Are you considering
an SDN based run-time dynamic reconfiguration?
OA5: No. We rather consider physical topologies in a conceptual model.
However, as maintaining the foundation for network connectivity,
our approach may also contribute to path control in SDN.
So, we mention "We focus on the connectivity at the most
fundamental level in many network systems for not only
communication but also transportation, power-supply, and other
infrastructures, while our approach may be useful for path control
or failure detection e.g. by software-defined network based
reconfiguration on communication systems with switches in managing
reliability, latency, or security at some service levels [20][21][22]."
in lines 106-110 on page 3.

RC6: Can you present the algorithms in the form of short flow chart
of small steps or an algorithm?
OA6: We add the list of steps in lines 87-99 on page 3.

RC7: The citations can be improved. Authors can consider the citations
in the area of reliability and SDN such as:
Song, Sejun, et al. "Control path management framework for enhancing
software-defined network (SDN) reliability." IEEE Transactions on
Network and Service Management 14.2 (2017): 302-316.
Hu, Yannan, et al. "Reliability-aware controller placement for
software-defined networks." 2013 IFIP/IEEE International Symposium
on Integrated Network Management (IM 2013). IEEE, 2013.
Thyagaturu, Akhilesh S., et al. "Software defined optical networks
(SDONs): A comprehensive survey." IEEE Communications Surveys &
Tutorials 18.4 (2016): 2738-2786.
OA7: We add these papers to reference and cite them in line 110 on page 3.